# Visual complexity of dental intake forms and its association with dental treatment outcomes: A retrospective cohort study

Yojiro Umezaki[1]*, Takeaki Sudo[2], Haruhiko Motomura[1,3], Hiromitsu Morita[4], Toru Naito[1]

1 Section of Geriatric Dentistry, Department of General Dentistry, Fukuoka Dental College, Fukuoka, Japan, 2 Department of Educational Media Development, Graduate School of Medical and Dental Sciences, Institute of Science, Tokyo, Japan, 3 Japan Imaging Center of Psychiatry and Neurology, Fukuoka, Japan, 4 Section of Dentistry for the Disabled, Department of Oral Growth and Development, Fukuoka Dental College, Fukuoka, Japan

* umezaki@fdcnet.ac.jp

## Abstract

### Background

While many factors influence dental treatment outcomes, the visual characteristics of intake forms—such as the amount of handwriting—remain largely unexplored. Clinical impressions suggest that minimal or excessive form completion may reflect patient engagement or psychological disposition. To examine whether the visual complexity of intake forms, quantified as a "writing ratio," is associated with treatment prognosis in dental settings.

### Methods

This retrospective cohort study included 813 patients who received a comprehensive dental consultation at Fukuoka Dental College Hospital in 2016. Intake forms were scanned and processed using Python and OpenCV to calculate the writing ratio, defined as the percentage of black pixels in the image. Patients were categorized into tertiles (Low, Mid, High) based on this ratio. Multivariable logistic regression was used to assess associations with poor treatment outcomes (defined as drop-out or clinician-initiated discontinuation), adjusting for age, sex, diagnosis, and the experience of the most senior attending dentist. An exploratory scoring system was constructed using key predictors and evaluated via ROC analysis.

### Results

Patients in the Low writing ratio group had a significantly higher risk of poor outcomes compared to the Mid group (adjusted OR = 1.53, 95% CI: 1.07–2.18, p = 0.019). No significant difference was observed between the Mid and High groups. A subgroup

**Data availability statement:** The data underlying the results presented in this study are available from the Ethics Committee of Fukuoka Dental College (contact via yukuhiro@fdcnet.ac.jp) or from the corresponding author (umezaki@fdcnet.ac.jp) for researchers who meet the criteria for access to confidential data. The data cannot be shared publicly because they contain potentially identifying or sensitive patient information.

**Funding:** This study was supported in part by the JSPS KAKENHI grant No. 25K13428. There was no additional external funding received for this study.

**Competing interests:** The authors have declared that no competing interests exist.

of middle-aged females exhibited the highest dropout rate. The exploratory scoring system showed modest discriminative performance (AUC = 0.544).

## Conclusions

Lower visual complexity of intake forms may reflect disengagement or unclear communication and is associated with poorer treatment outcomes. Intake form appearance may serve as an early behavioral indicator and support risk stratification in dental care. Redesigning intake forms to capture both structural and behavioral cues may enhance early clinical assessment and care planning.

---

## Introduction

Dental treatment outcomes are influenced by a complex interplay of factors. A possible framework is to classify these factors into provider-side [1–3] and patient-side [4–6] variables. Provider-related factors -such as the technical skill of the dentist, the competence of auxiliary staff, the availability of appropriate instruments, and the quality of the clinical environment- are well recognized as affecting treatment success. Patient-related elements may include medical history, oral condition, personality traits, and level of cooperation. The interaction between provider-side and patient-side factors—particularly in terms of communication, trust, and the quality of the patient-dentist relationship—is also crucial for successful treatment outcomes. [7].

However, another useful perspective is to divide these factors based on whether they can be addressed through provider-side efforts, such as training, clinical experience, or institutional adjustments. From this view, many factors including the difficulty of the procedure or the presence of systemic disease can be anticipated and managed to some extent through appropriate planning and response. In contrast, patient characteristics such as vague complaints, low engagement, or behavioral tendencies often lack clear criteria for identification and are not easily resolved through clinical skill alone. These traits are typically judged through intuition rather than formal assessment [5,8].

Intake forms in dental clinics are typically semi-structured, offering checkboxes for symptoms and medical history items. In many cases, a patient can complete the form simply by checking predefined options, resulting in a clean and easily interpretable document. Such forms often correspond to relatively straightforward cases, in which the chief complaint is readily identifiable and treatment planning proceeds without complication. In contrast, forms that are filled in with excessive handwriting -often in margins or blank spaces- can introduce ambiguity. When patients write extensively, or in an unstructured manner, it may obscure the central complaint, complicate diagnostic reasoning, and delay or disrupt the treatment process. These "messy" forms may reflect a complex or unresolved agenda. While this has not been empirically demonstrated, such forms often give the impression of ambiguity or difficulty during clinical interviews and may contribute to treatment discontinuation, unsatisfactory outcomes, or even medico-legal misunderstandings.

Based on these clinical observations, we hypothesized that the visual and structural characteristics of intake forms may reflect underlying patient factors such as cooperation, communicative style, or psychological disposition. Specifically, we asked whether the appearance of a patient's intake form -quantified through a writing ratio- could be used to predict treatment prognosis. The null hypothesis was that there is no association between the visual characteristics of the intake form and dental treatment prognosis. This perspective is important because it addresses an overlooked aspect of patient behavior that may be visible even before clinical interaction begins. By examining the visual structure of intake forms, we aim to fill a gap in current dental risk assessment and explore whether this simple indicator can support early identification of at-risk patients and provide insights to improve the design of future intake forms.

## Materials and methods

### Study design and participants

This retrospective cohort study was conducted using data from patients who underwent their initial comprehensive dental consultation at Fukuoka Dental College hospital during the 2016 fiscal year. Patients were eligible if both medical records and scanned intake forms were available. Exclusion criteria included patients whose first visit occurred outside of regular clinic hours or those with only a single recorded visit, as reliable assessment of prognosis was not feasible in these cases.

### Power calculation and preliminary analysis

The preliminary analysis quantified the amount of writing on intake forms by applying Optical Character Recognition (OCR) to the scanned images and counting the number of extracted characters. This approach served as an initial proxy for written content volume, although it was later replaced by a more objective image-based measure due to variability in OCR accuracy across handwriting styles. In this preliminary dataset, a higher character count was associated with poorer prognosis, aligning with our initial hypothesis that more complex or verbose intake forms may reflect more challenging clinical scenarios. Based on these preliminary results, we performed a sample size calculation. Using a convenience sample of 33 patients and an observed effect size (Cohen's d = 0.5), a two-tailed power analysis (alpha = 0.05, power = 0.8) indicated that a minimum of 128 patients (64 per group) would be required. The final dataset exceeded this threshold, confirming sufficient statistical power for the main analysis. For the full analysis, given the inconsistent accuracy of OCR across different handwriting styles, we opted to use a more robust and objective measure: the image-based writing ratio.

### Data collection

Data extraction and image analysis were performed in October 2022. Patient demographics (age and sex), diagnoses (multiple entries allowed), number of visits, treatment outcomes, and information regarding attending dentists (including the number of attending dentists and their years of experience) were extracted from electronic records. Diagnostic categories were defined as periodontal, restorative (including direct restorations and endodontic treatment), prosthodontic, oral surgery, orthodontic, and others. The others category included conditions such as dental phobia, burning mouth syndrome, atypical odontalgia, and neuropathic pain. In this study, the "dental phobia" category encompassed not only dental treatment anxiety but also related conditions such as abnormal choking reflex and exaggerated gag reflex. Diagnoses were not mutually exclusive; patients could be assigned to more than one diagnostic category based on their clinical presentation. Scanned questionnaire images were processed using a custom Python-based algorithm. The writing ratio was defined as the percentage of black pixels (representing intake form itself and handwriting) relative to the total pixel count of the form. This writing ratio serves as a quantitative proxy for the visual complexity of the intake form — capturing how much content, whether typed or handwritten, appears on the form regardless of its semantic content.

### Image processing and quantification of questionnaire filling area

To quantify the amount of writing on the patient questionnaires, all image processing tasks were performed using Python (Python Software Foundation; Wilmington, DE, USA) version 3.9 and the OpenCV library (OpenCV.org; Palo Alto, CA, USA) version 4.5.5. Each scanned or photographed questionnaire image was first converted to grayscale, followed by binarization using Otsu's thresholding method (cv2.threshold). The number of black pixels was then calculated using cv2.countNonZero and subtracted from the total pixel count to estimate the amount of written content, represented as a proportion of all pixels. All images were processed under identical conditions. A detailed algorithm is provided in S1 File.

### Outcome definition

Treatment outcomes were classified into four categories: "Cured", "Ongoing", "Dropped out", and "Clinician-initiated discontinuation".

- "Cured" refers to cases in which the chart clearly documented resolution of the chief complaint and formal completion of treatment.
- "Ongoing" refers to cases in which treatment was still in progress at the time of data collection.
- "Dropped out" refers to patients who failed to return for a scheduled visit and whose treatment was thus interrupted without resolution.
- "Clinician-initiated discontinuation" refers to cases in which treatment was discontinued by the provider despite the chief complaint not being resolved.

For statistical analyses, a poor outcome was defined as either "Dropped out" or "Clinician-initiated discontinuation."

### Statistical analysis

The writing ratio was analyzed both continuously and categorically, using tertiles and percentile cutoffs. Group differences in outcome distributions were assessed using chi-squared test and Fisher's exact test, as appropriate. Multivariable logistic regression was employed to investigate associations between writing ratio and poor outcomes, adjusting for potential confounders: age (categorized as <50, 50–64, and ≥65 years), sex, diagnosis, and the most experienced attending dentist (in years). An exploratory ROC analysis was conducted to evaluate the performance of a composite scoring system. All statistical analyses were conducted using SPSS Statistics (version 29.0.1.0; IBM, Armonk, NY, USA). A two-sided p-value < 0.05 was considered statistically significant. This study was conducted in accordance with the Declaration of Helsinki and was approved by the Ethics Committee of Fukuoka Dental College (approval no. 541). Informed consent was waived by the committee due to the retrospective design and use of de-identified data.

## Results

### Participant characteristics

A total of 1,142 patients who had a first-time comprehensive dental visit in 2016 were initially considered. From these, 97 cases involving initial visits conducted outside of regular clinic hours and 269 one-time-visit cases were identified, with 37 patients overlapping in both groups. The remaining 813 patients formed the main analysis cohort. The mean age of the cohort was 49.2 years (SD = 17.1), ranging from 13 to 99 years, and 53.6% were male. The most experienced attending dentist for each case had an average of 12.78 years of clinical experience (SD = 8.72), with a range from 0 to 37 years. The writing ratio ranged from 5.4612% to 11.6074%, with a slightly right-skewed distribution (Fig 1). Table 1 summarized the demographic and diagnostic characteristics of the patients included in the main analysis cohort. Patients were

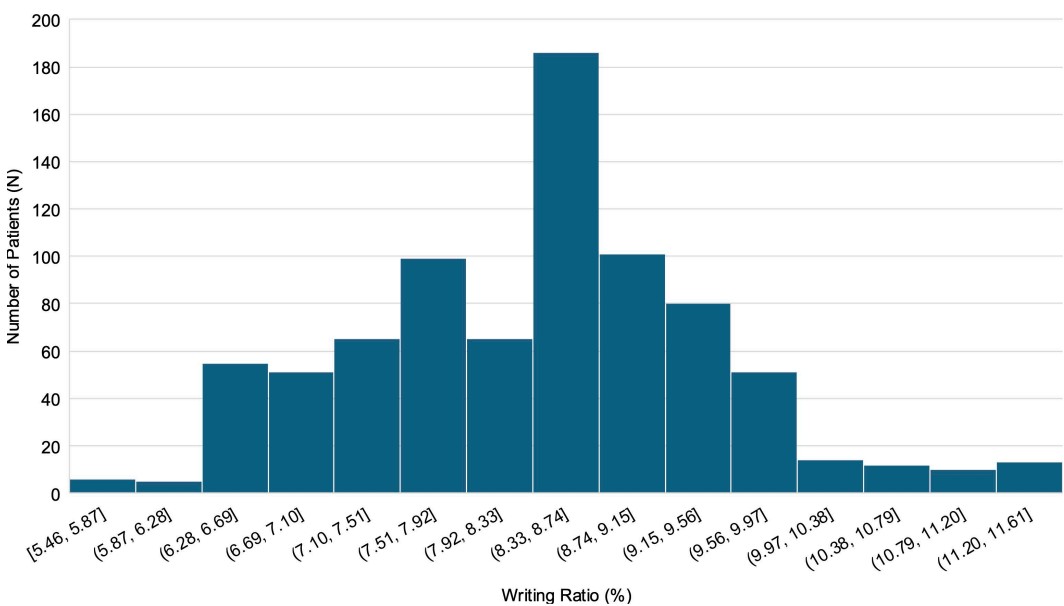

**Fig 1. Distribution of writing ratio across the study cohort.** This histogram illustrates the distribution of writing ratio, defined as the percentage of black pixels on the intake form image. The cohort was divided into tertiles based on this measure: Low (<7.8900%, n = 271), Mid (≥7.8900% and <8.7626%, n = 270), and High (≥8.7626%, n = 272).

**Table 1. Participant characteristics by writing ratio group.**

| | | Total | Low Ratio (< 7.8900%) | Mid Ratio (≥7.8900% and <8.7626%) | High Ratio (≥8.7626%) |
|---|---|---|---|---|---|
| Number of patients | | 813 | 271 | 270 | 272 |
| Age (mean ± SD) | | 56.11 ± 19.26 | 56.24 ± 19.12 | 55.56 ± 19.51 | 56.53 ± 19.22 |
| Male: Female | | 334: 479 | 104: 167 | 120: 150 | 110: 162 |
| Writing Ratio (mean ± SD) | | 8.38 ± 1.10 | 7.16 ± 0.52 | 8.45 ± 0.23 | 9.54 ± 0.68 |
| Treatment outcome | Cured (%) | 36.8 | 32.8 | 39.3 | 38.2 |
| | Ongoing (%) | 2.3 | 1.1 | 4.1 | 1.8 |
| | Dropped out (%) | 57.7 | 63.8 | 50.4 | 58.8 |
| | Clinician-initiated discontinuation (%) | 3.2 | 2.2 | 6.3 | 1.1 |
| Diagnosis | Periodontal (%) | 88.1 | 87.8 | 95.9 | 80.5 |
| | Restorative (%) | 68.6 | 71.2 | 64.4 | 70.2 |
| | Prosthetic (%) | 31.1 | 31.4 | 29.6 | 32.4 |
| | Oral surgical (%) | 16.9 | 18.5 | 14.8 | 17.3 |
| | Orthodontic (%) | 0.7 | 1.1 | 0.7 | 0.4 |
| | Others (%) | 3.6 | 4.4 | 3.0 | 3.3 |
| Years of experience of the most senior attending dentist (mean ± SD) | | 12.78 ± 8.72 | 13.71 ± 8.58 | 11.80 ± 8.74 | 12.82 ± 8.79 |

"Writing Ratio" refers to the percentage of black pixels (representing text and form elements) relative to the total number of pixels in the scanned intake form image. Higher values indicate greater visual complexity or amount of written content.

grouped into tertiles based on the writing ratio: Low (< 7.8900%, n = 271), Mid (≥7.8900 and <8.7626%, n = 270), and High (≥ 8.7626%, n = 272). Representative examples of intake forms from each writing ratio group are provided in S1 Fig.

## Association between ratio and prognosis

The proportion of poor outcomes (i.e., treatment dropout or clinician-initiated discontinuation) by writing ratio group is shown in **Fig 2**. Patients in the Low Ratio group had a significantly higher rate of poor prognosis compared to the Mid and High groups (p < 0.028, chi-square test). To further explore this association, we conducted a multivariable logistic regression analysis for poor treatment outcomes (**Table 2**). Variables included in the model were writing ratio category, age category (< 50, 50–64 and ≥ 65 years), sex, and all diagnosis categories (periodontal, restorative, prosthetic, oral surgery, orthodontic, and others), along with the years of experience of the most experienced attending dentist. The Mid group was used as the reference category to allow balanced comparisons between low and high writing ratio groups. Compared to the Mid group, the Low Ratio group had an adjusted odds ratio of 1.53 (95% CI: 1.07–2.18, p = 0.019, logistic regression). No significant difference was found between the Mid and High Ratio groups (p = 0.257, logistic regression). Among the diagnostic categories included as covariates, only periodontal diagnosis showed a statistically significant association with poor outcome.

## Subgroup trends in prognosis by age and sex

While the multivariable logistic regression model did not identify age or sex as significant predictors when entered as independent variables, we hypothesized that non-linear patterns or interaction effects might exist. To investigate this possibility, we conducted subgroup analyses based on combined age and sex categories. Specifically, we examined poor outcome rates (i.e., treatment dropout or clinician-initiated discontinuation) by age and sex to determine whether specific subgroups exhibited distinct prognostic trends. Interestingly, middle-aged female patients (defined as those aged 50–64) exhibited the highest rate of poor outcomes, with 67% (73 out of 109) meeting the criteria for either dropout or

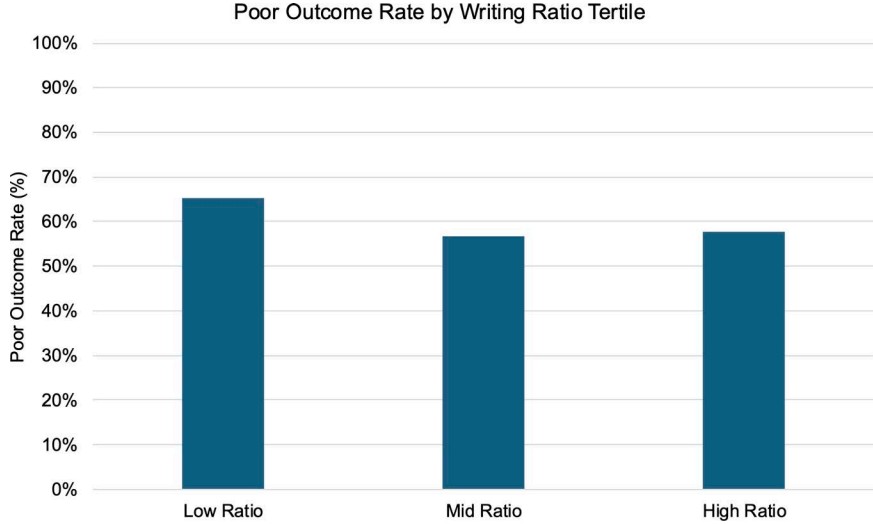

**Fig 2. Proportion of poor outcomes by writing ratio group.** Bar graph shows the proportion of patients with each treatment outcome within the Low (<7.8900%), Mid (≥7.8900 and <8.7626%), and High (≥8.7626%) writing ratio groups. The writing ratio quantifies the amount of visible content on the intake form (including handwriting and checkboxes) and is used as a proxy for patient engagement during the first visit. Y-axis represents percentage (0–100%). The proportion of patients who experienced poor outcomes (dropout or clinician-initiated discontinuation) was highest in the Low Ratio group, supporting the association between minimal intake form completion and treatment discontinuation.

**Table 2. Multivariable logistic regression analysis of factors associated with poor treatment outcomes.**

| Variable | Adjusted OR | 95% CI | p-value |
|---|---|---|---|
| **Writing Ratio Category (Reference: Mid)** | | | |
| Low Ratio | 1.53 | 1.07 - 2.18 | 0.019* |
| High Ratio | 1.23 | 0.86 - 1.74 | 0.257 |
| **Age Category (Reference: < 50 years)** | | | |
| 50 - 64 years | 1.28 | 0.86 - 1.91 | 0.222 |
| ≥65 years | 1.08 | 0.76 - 1.52 | 0.682 |
| Male (vs Female) | 0.93 | 0.70 - 1.25 | 0.632 |
| **Diagnosis** | | | |
| Periodontal | 1.64 | 1.06 - 2.55 | 0.028* |
| Restorative | 0.98 | 0.71 - 1.34 | 0.884 |
| Prosthetic | 1.06 | 0.76 - 1.47 | 0.737 |
| Oral surgical | 1.17 | 0.79 - 1.74 | 0.428 |
| Orthodontic | 0.50 | 0.10 - 2.65 | 0.417 |
| Others | 1.67 | 0.99 - 1.02 | 0.235 |
| Experience of most senior dentist (in years) | 1.00 | 0.99 - 1.02 | 0.895 |

All variables included in the logistic regression model—writing ratio (categorical), age group, sex, diagnosis category, and the experience level of the most senior attending dentist—are reported, regardless of statistical significance.

discontinued treatment without resolution (Table 3). This is notably higher than the overall poor outcome rate of 60% in the full cohort. This finding suggests that age–sex interactions may influence prognosis in ways not fully captured by individual variables in multivariable models. Although the observed difference did not reach statistical significance (p = 0.160, chi-square test), the relatively high poor outcome rate in this subgroup may suggest a trend worth exploring further in larger or more targeted datasets.

## Supplementary analysis: Characteristics of patients with extremely high writing ratios

Although patients in the High writing ratio group did not show a statistically significant increase in poor outcomes (i.e., treatment dropout or clinician-initiated discontinuation) compared to the Mid group, we explored whether patients with extremely high writing ratios exhibited any clinical tendencies. Among the top 10% of patients based on writing ratio (n = 81), 3.7% (3/81) had a phobia-related diagnosis. In contrast, only 1.0% (7/732) of the remaining patients were similarly diagnosed (Fig 3). While this difference did not reach conventional statistical significance due to the small number of cases, Fisher's exact test indicated a marginal trend (p = 0.068, Fisher's exact test). The observed distribution suggests

**Table 3. Poor outcome rate by age-sex subgroup.**

| Age group | Sex | Poor outcome (n) | Total (n) | Poor outcome rate (%) |
|---|---|---|---|---|
| < 50 | Male | 80 | 126 | 63.49 |
| < 50 | Female | 87 | 161 | 54.04 |
| 50 - 64 | Male | 40 | 66 | 60.61 |
| 50 - 64 | Female | 74 | 109 | 67.89 |
| ≥ 65 | Male | 80 | 142 | 56.34 |
| ≥ 65 | Female | 134 | 209 | 64.11 |

Age group "<50" includes participants aged 13–49 years. The minimum age in the study cohort was 13 years.

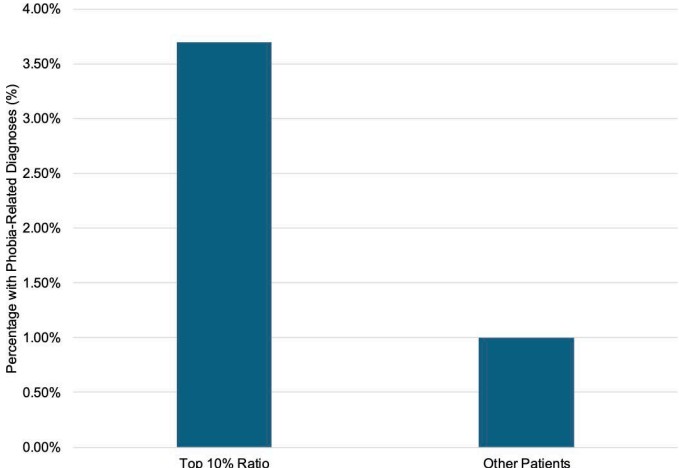

**Fig 3. Proportion of patients with phobia-related diagnoses in the highest writing ratio group.** The bar graph compares the proportion of patients diagnosed with phobia-related conditions between those in the top 10% of writing ratio (n = 81) and the remainder of the cohort (n = 732). Although not statistically significant, phobia-related diagnoses were more frequently observed among patients with exceptionally high writing ratios, suggesting a potential association between excessive intake form completion and heightened anxiety or vigilance.

a potential association between excessive form completion and heightened anxiety or vigilance, consistent with clinical impressions.

## Exploratory scoring system

To explore the clinical applicability of the findings, we constructed a simple scoring system based on the key predictors of poor outcome (i.e., treatment dropout or clinician-initiated discontinuation) identified through multivariable and subgroup analyses. This system was not intended as a predictive model, but rather as a conceptual tool to highlight combinations of factors that may warrant closer attention in early clinical encounters.

The scoring system included the following elements:

- +2 points for having a low writing ratio (lowest tertile of the cohort), which was significantly associated with poor outcomes in both univariate and multivariable analyses.

- +1 point for having a diagnosis in the periodontal category, which was the only diagnostic group significantly associated with poor outcome in the regression model.

- +1 point for being a middle-aged female (50–64 years), based on subgroup analysis showing the highest poor outcome rate (67%) among all age–sex categories. Although this result did not reach conventional statistical significance (p = 0.116), the observed pattern suggested a potential trend worth further exploration.

Patients with a total score of 3 or more were classified as high-risk. To examine the performance of this threshold, we conducted a ROC analysis using the binary score. **Fig 4** shows the ROC curve generated using the dichotomized scoring system. The area under the curve (AUC) was 0.544 (p = 0.032, ROC analysis), with a sensitivity of 33.3% and specificity of 75.5% (Youden Index = 0.088). These results suggest that while the score captures some aspects of risk, its predictive power remains modest, indicating that such a simple classification may not adequately reflect the complexity of real-world treatment outcomes.

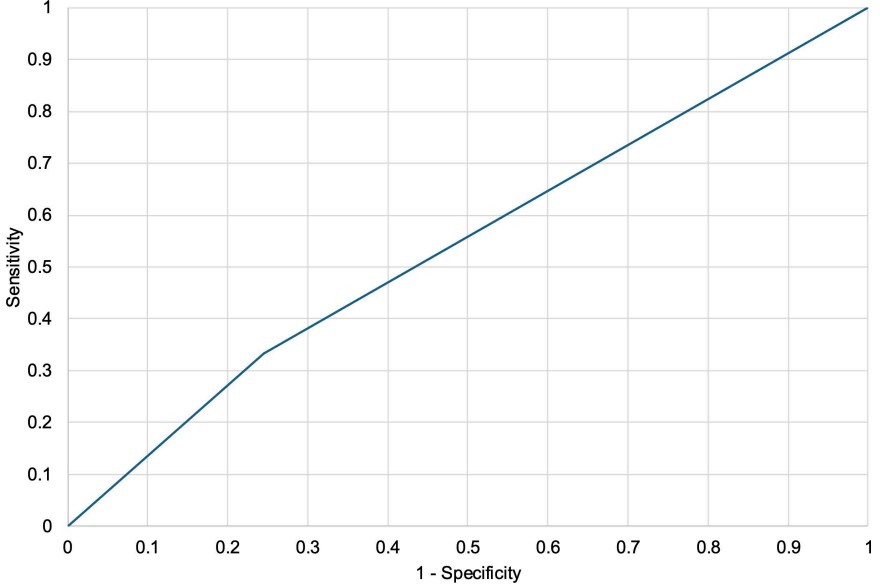

**Fig 4. ROC curve of the exploratory scoring system.** Receiver operating characteristic (ROC) curve evaluating the scoring system's ability to identify patients at risk of poor outcomes. Using a cutoff of ≥3 points, the area under the curve (AUC) was 0.544 (p = 0.032), with sensitivity of 33.3%, specificity of 75.5%, and a Youden Index of 0.088, indicating modest discriminative performance.

## Discussion

This study was designed to test the null hypothesis that the visual complexity of dental intake forms—quantified by the writing ratio—has no association with treatment outcomes. We focused on whether this structural characteristic, independent of content itself, could serve as an early behavioral signal related to prognosis.

Contrary to this hypothesis, our findings demonstrated that patients who completed intake forms with minimal written content had significantly higher rates of poor treatment outcomes (i.e., treatment dropout or clinician-initiated discontinuation). This suggests that disengagement or lack of clarity at the initial visit may act as early warning signs for poor outcomes.

While some previous studies have used natural language processing or analyzed the content of free-text responses in intake forms [9,10], few have examined the visual or structural characteristics. Our findings indicate that even simple visual metrics—such as the proportion of space filled in—may contain clinically meaningful signals. Interestingly, patients who wrote more extensively were less likely to have poor outcomes, though anxiety-related diagnoses were slightly more frequent in that group. This distinction suggests at least two types of complex patients: those who under-express and disengage, and those who over-express, potentially reflecting psychological distress. Each type may require different clinical approaches and strategies for early intervention.

Previous research has identified various predictors of dental treatment dropout, including patient-side factors such as age [11–16], sex [11–13,17], distance traveled [16,17], socioeconomic status [6,11–13,17,18], family structure [18], personality traits [19], and insurance status [15,16], and provider-side factors including clinician experience [16]. These factors are undoubtedly important, but our dataset did not permit detailed analysis of many of these variables. Rather than replicating previous investigations into sociodemographic or provider-level predictors, our study adopted a different perspective—focusing on a behavioral and structural indicator: the visual complexity of the intake form. This novel angle offers a complementary approach to conventional poor outcome risk models and may contribute to a more nuanced understanding of patient engagement, particularly in the early stages of dental care.

It is also notable that, across the cohort, the overall proportion of poor outcomes was relatively high, ranging from 50% to 60%. While previous studies have typically reported dropout rates in general dental care ranging from 30.0% to 32.6% [11,12], our study observed a notably higher proportion. This apparent discrepancy may be partly explained by our exclusion of patients whose initial visits occurred outside of regular clinic hours and simple one-time visits, which likely resulted in a cohort enriched for patients requiring ongoing treatment. From a clinical standpoint, the observed dropout rate aligns with real-world impressions, particularly in settings involving comprehensive care or multidisciplinary consultation. Therefore, although numerically high, the dropout rate reported here may represent a realistic depiction of challenges faced in maintaining treatment continuity in complex dental cases. This finding highlights the need for better early identification of at-risk patients and suggests that the existing intake processes may not be sufficiently sensitive to underlying risks that affect treatment continuity and success.

We also found that patients with a diagnosis in the periodontal category were more likely to have poor outcomes. This may reflect the nature of periodontal treatment itself, which often prioritizes long-term maintenance and disease control over complete resolution or "cure" [20,21]. As a result, such cases may be less likely to meet the criteria for favorable outcomes in retrospective evaluations that use standardized classification schemes, despite being appropriately managed over time. This highlights the importance of considering diagnostic context when interpreting prognostic signals.

To explore the practical implications of our findings, we proposed a simple scoring system based on intake form complexity, diagnosis category, and age-sex demographics. However, the performance of this scoring system was modest, with an AUC of 0.542 and low sensitivity (33.3%) at the 3-point cutoff. These results underscore the limitations of relying solely on basic intake information to predict clinical trajectories. While certain visual and demographic signals can provide clues, they are insufficient for comprehensive risk stratification.

Importantly, the limitations of the exploratory scoring system highlight a broader issue: traditional intake forms, designed primarily for record-keeping, may no longer suffice in a clinical environment that demands early and dynamic risk identification. In several medical fields, the necessity and utility of standardized intake forms have already been discussed and emphasized [22–24]. However, such discourse has yet to gain traction in dentistry. Given the complexity and diversity of dental patients, the need for a unified, structured intake form in dental practice is beyond question. Future efforts should focus on designing more structured, behaviorally informative intake forms that capture not only symptoms and history but also aspects of psychological readiness, communication style, and engagement. By integrating such elements, intake forms could transform into active clinical tools that support early intervention, personalized care planning, and ultimately, improved treatment outcomes. Beyond clinical implications, our findings may also contribute to educational practices in dentistry. Teaching students and clinicians to recognize visual cues in patient intake behavior—such as limited written responses—could enhance their ability to detect early signs of disengagement or psychological burden. Integrating this perspective into training may promote more attentive patient evaluation from the very first visit, thereby improving risk assessment and communication strategies.

This study has several strengths. First, it introduces a novel and objective metric—the writing ratio—to quantify the visual complexity of dental intake forms, offering a unique lens through which to understand early patient engagement. Second, the use of image-based analysis allowed for standardized and reproducible quantification across a large sample. Third, by applying multivariable regression and subgroup analyses, we were able to control for key confounding factors such as age, sex, and diagnosis, thereby strengthening the robustness of our findings.

This study also has several limitations. First, the retrospective design restricts the ability to infer causal relationships, and unmeasured confounders—such as education level, socioeconomic status, or psychological background—may have influenced the observed associations. Second, while the writing ratio provides an objective measure of visual complexity, it does not account for the semantic content or coherence of what is written. Some patients may write concisely and effectively, while others may use excessive but unclear language—both resulting in similar ratios. Third, although the sample size was sufficient for primary analyses, subgroup findings (e.g., age–sex combinations or specific diagnoses) should be

interpreted with caution due to limited statistical power. Finally, this was a single-center study conducted in a college hospital setting, which may limit the generalizability of results to other practice environments.

## Conclusion

This study suggested that the visual complexity of dental intake forms—measured by a simple writing ratio—may reflect early behavioral signals relevant to treatment prognosis. Intake forms with minimal written content were associated with a higher risk of poor outcomes, indicating such structural features could help identify patients who may require additional attention or support. These findings highlight the potential utility of intake form design not only for documentation, but also as a tool for early risk detection and personalized care planning in dental practice.

## Supporting information

**S1 File. Python code used for image-based calculation of writing ratio.** This script performs image processing to quantify the writing ratio of scanned or photographed patient intake forms. Implemented in Python (v3.9) with the OpenCV library (v4.5.5), the script converts each image to grayscale and applies binarization using Otsu's thresholding method. The number of black pixels—representing printed elements and handwritten text—is calculated by subtracting the count of non-zero pixels from the total pixel count. The writing ratio is then computed as the proportion of black pixels relative to all pixels in the image. All images were processed under identical conditions to ensure consistency.
(PDF)

**S1 Fig. Representative examples of scanned intake forms.** (A) Low writing ratio (5.4612%); (B) Mid writing ratio (8.4545%); (C) High writing ratio (10.5680%). These examples are scanned images of Japanese-language dental intake forms used in the present study. Each form includes structured items with checkboxes (e.g., current symptoms, medical history) and optional free-text sections. The amount of handwriting varies across groups, reflecting differences in patient engagement or communicative style. Personal information has been redacted.
(TIFF)

## Author contributions

**Conceptualization:** Yojiro Umezaki.

**Data curation:** Yojiro Umezaki, Haruhiko Motomura.

**Formal analysis:** Yojiro Umezaki, Hiromitsu Morita.

**Funding acquisition:** Toru Naito.

**Investigation:** Yojiro Umezaki, Haruhiko Motomura.

**Methodology:** Yojiro Umezaki, Takeaki Sudo.

**Project administration:** Haruhiko Motomura.

**Resources:** Yojiro Umezaki.

**Software:** Yojiro Umezaki, Takeaki Sudo.

**Supervision:** Toru Naito.

**Validation:** Hiromitsu Morita.

**Visualization:** Yojiro Umezaki.

**Writing – original draft:** Yojiro Umezaki.

**Writing – review & editing:** Takeaki Sudo, Haruhiko Motomura, Hiromitsu Morita, Toru Naito.

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
