## [Decision Letter · Decision Letter 0]

15 Jul 2025

Dear Dr. Umezaki,

Thank you for submitting your manuscript to PLOS ONE. After careful consideration, we feel that it has merit but does not fully meet PLOS ONE’s publication criteria as it currently stands. Therefore, we invite you to submit a revised version of the manuscript that addresses the points raised during the review process.

**Please note that our external Reviewer has not been satisfied with your draft, and has recommended numerous revisions (please see comments added below). Therefore, I have double-checked your manuscript, and have commented on your submitted version (please see comments referring to Reviewer #1 as given below). In total, your manuscript would not seem acceptable in its present form. I have decided to wait for your revised version, to take a second view on your draft. Please stick to each and every comment, and note that re-review will be mandatory.**

We look forward to receiving your revised manuscript.

Kind regards,

Andrej M Kielbassa

Academic Editor

PLOS ONE

**Journal Requirements:**

1. When submitting your revision, we need you to address these additional requirements. Please ensure that your manuscript meets PLOS ONE's style requirements, including those for file naming. The PLOS ONE style templates can be found at https://journals.plos.org/plosone/s/file?id=wjVg/PLOSOne_formatting_sample_main_body.pdf and https://journals.plos.org/plosone/s/file?id=ba62/PLOSOne_formatting_sample_title_authors_affiliations.pdf 2. Thank you for stating in your Funding Statement: This study was supported in part by the JSPS KAKENHI grant No. 25K13428.  Please provide an amended statement that declares *all* the funding or sources of support (whether external or internal to your organization) received during this study, as detailed online in our guide for authors at http://journals.plos.org/plosone/s/submit-now.  Please also include the statement “There was no additional external funding received for this study.” in your updated Funding Statement. Please include your amended Funding Statement within your cover letter. We will change the online submission form on your behalf. 3. We note that you have indicated that there are restrictions to data sharing for this study. For studies involving human research participant data or other sensitive data, we encourage authors to share de-identified or anonymized data. However, when data cannot be publicly shared for ethical reasons, we allow authors to make their data sets available upon request. For information on unacceptable data access restrictions, please see http://journals.plos.org/plosone/s/data-availability#loc-unacceptable-data-access-restrictions.  Before we proceed with your manuscript, please address the following prompts: a) If there are ethical or legal restrictions on sharing a de-identified data set, please explain them in detail (e.g., data contain potentially identifying or sensitive patient information, data are owned by a third-party organization, etc.) and who has imposed them (e.g., a Research Ethics Committee or Institutional Review Board, etc.). Please also provide contact information for a data access committee, ethics committee, or other institutional body to which data requests may be sent. b) If there are no restrictions, please upload the minimal anonymized data set necessary to replicate your study findings to a stable, public repository and provide us with the relevant URLs, DOIs, or accession numbers. Please see http://www.bmj.com/content/340/bmj.c181.long for guidelines on how to de-identify and prepare clinical data for publication. For a list of recommended repositories, please see https://journals.plos.org/plosone/s/recommended-repositories. You also have the option of uploading the data as Supporting Information files, but we would recommend depositing data directly to a data repository if possible. Please update your Data Availability statement in the submission form accordingly. 4. Please upload a new copy of Figures 1 to 4 as the detail is not clear. Please follow the link for more information: https://blogs.plos.org/plos/2019/06/looking-good-tips-for-creating-your-plos-figures-graphics/" https://blogs.plos.org/plos/2019/06/looking-good-tips-for-creating-your-plos-figures-graphics/ 5. Please include captions for your Supporting Information files at the end of your manuscript, and update any in-text citations to match accordingly. Please see our Supporting Information guidelines for more information: http://journals.plos.org/plosone/s/supporting-information.

Reviewers' comments:

Reviewer's Responses to Questions

**Comments to the Author**

1. Is the manuscript technically sound, and do the data support the conclusions?

Reviewer #1: No

Reviewer #2: Partly

2. Has the statistical analysis been performed appropriately and rigorously?

Reviewer #1: No

Reviewer #2: Yes

3. Have the authors made all data underlying the findings in their manuscript fully available?

Reviewer #1: No

Reviewer #2: No

4. Is the manuscript presented in an intelligible fashion and written in standard English?

Reviewer #1: Yes

Reviewer #2: No

**Reviewer #1:**  This would seem an interesting paper, covering some new aspects. Please see comments given below, and revise carefully.

As a general comment, please study the Journal's Guidelines for Authors. Remember that your draft must be fully adapted to Journal style to be eligible to any proceeding.

Abstract

- With 244 words, this section does not provide maximum information. Remember that 300 words are allowed for with this section, so please add as much information as possible here. Please note that this section is most considered important to decide for switching to your full text.

- With your Conclusions, please stick exclusively to your revised aims. Do not simply repeat your results here. Do not speculate. Instead, provide a reasonable and generalizable extension of your outcome.

Intro

- Please revise for sound spacebar use, see "(...) provider-side(1-3) and patient67 side(4-6) (...).". Again, stick to uniform Journal style, and revise thoroughly.

- What about a sound rationale? Please revise carefully.

- "Specifically, we asked whether the appearance of a patient's intake form -quantified (...)." What about a sound and valid null hypothesis?

Meths

- Details or references must be provided with your "applicable institutional guidelines" and "regulatory requirements".

- "Python (v3.9) and the OpenCV (v4.5.5)" - more details must be given here. With ALL materials and methodologies (including statistical software), please use general names with your text, followed by (brand name; manufacturer, city, ST[ate - abbreviated, if US], country) in parentheses. Stick to semicolon. Remember that reproducibility must be ensured.

- Do not use legal terms with your full text. Delete "Inc.", "Corp.", and so on.

Results

- Please add statistical tests with your p values.

- Provide representative "photographed questionnaire images".

- The readers surely would like know what you mean when referring to "specifically the writing ratio".

Disc

- Stick to H0 when starting this section.

- What about the strengths of your study?

- What about the limitations?

- What about answers to your teaching objective?

Concl

- Again, with your Conclusions, please stick exclusively to your revised aims. Do not simply repeat your results here. Do not speculate. Instead, provide a reasonable and generalizable extension of your outcome.

Refs

- Please adapt thoroughly to uniform Journal style.

**Reviewer #2:**  Overall Comments:

The authors have done a good job overall, but the manuscript needs major revisions to improve clarity and flow. Some parts are confusing or feel disconnected. It would help to rewrite the paper with a clearer structure and storyline. The analysis section especially needs more explanation — it should clearly state what was analyzed and why.

Specific Comments:

1. Lines 71–72: The sentence is unclear and should be rewritten for clarity.

2. Lines 82–93: These statements need to be supported by appropriate references.

3. Line 110: Please clarify this line.

4. Why was data extracted in 2022 for the 2016 fiscal year? Please explain the rationale.

5. Line 186: The statement that there are 288 patients in each group is incorrect. Please revise and ensure accuracy.

6. The mean age of participants is nearly identical across groups. Was this by chance, or was it an intentional aspect of the study design? Please clarify.

7. Writing Ratio (8.76%), please confirm whether this value relates to the high or medium writing ratio category. As written, it belongs to both.

8. Table 1: Consider adding variable names such as treatment outcomes and diagnoses, then the subcategories

9. Is diagnosis mutually exclusive? Can a single patient have more than one diagnosis? If so, please state this clearly in the manuscript.

10. In Table 1, please clarify what “Ratio (Mean ± SD)” refers to.

11. I strongly recommend including age as a categorical variable in the analysis, rather than reporting only the mean.

12. Line 188: This appears to be a figure title. Please begin the following sentence on a new line. Also, results from the figure should be described in the text. (Note: the figure is unclear and difficult to interpret in its current form.)

13. “Outcome Distribution in Excluded Cases.” This section is unnecessary and should be removed.

14. Figure 2: The Y-axis should range from 0 to 100%. Also, clearly define what is meant by low, medium, and high writing ratios.

15. Please consistently refer to and define what constitutes a poor outcome to maintain reader clarity.

16. It is sufficient to round to the values to the nearest hundredth, other than P values.

17. In table 2, please explain why the "mid" ratio group was chosen as the reference group in the regression model.

18. All variables included in the regression model should be reported, whether or not they were statistically significant.

19. Table 3: Clarify what "<50" refers to. Please mention the exact age of the youngest participant.

20. Consider including an interaction term between age and sex in the regression model. Again, reporting age categorically rather than as a mean would be more informative.

21. “Experience of Most Senior Dentist” This variable appears only in the regression model and is not introduced or discussed elsewhere. Values of this variable need to be reported before the regression model.

22. Title "Supplementary Findings": Consider rephrasing this title to better reflect the content.

23. "Although the overall writing ratio was not significantly associated with poor outcomes in the High Ratio group" — please clarify the meaning of this statement.

24. Please explain how you calculated the numbers 114 and 1028, given that the total sample size is 813. The sentence in question is:

25. "Among the top 10% of patients based on writing ratio (n = 114), 2.6% (3/114) had a phobia-related diagnosis. In contrast, only 0.8% (8/1028) of the remaining patients were similarly diagnosed."

26. I expected to find an example of the intake form to better understand what information patients are asked to write. Please consider including one.

27. Lines 297–301: There is repetition in these lines. Please revise for conciseness.

28. I would suggest replacing "not more likely" with "less likely."

29. It's essential to highlight that this study analyzes the amount of writing in the intake form, regardless of content quality or completeness. Some patients may write less but provide concise and complete information, while others may write more but omit key details. This should be explicitly stated in the manuscript.

30. Please include the limitations of your study in your discussion.

**Do you want your identity to be public for this peer review?** For information about this choice, including consent withdrawal, please see our Privacy Policy

Reviewer #1: No

Reviewer #2: No

---

## [Author Response · Author response to Decision Letter 1]

1 Aug 2025

We thank the reviewers and editors for their helpful and constructive feedback. Below is our point-by-point response to each comment.

Reviewer #1: This would seem an interesting paper, covering some new aspects. Please see comments given below, and revise carefully.

Thank you for your encouraging remarks. We agree that the visual complexity of intake forms is an underexplored area in dental research, and we appreciate the opportunity to share our findings. We have carefully reviewed and addressed all specific comments provided, and we hope that the revised manuscript reflects the novelty and potential clinical relevance of our approach.

As a general comment, please study the Journal's Guidelines for Authors. Remember that your draft must be fully adapted to Journal style to be eligible to any proceeding.

Abstract

- With 244 words, this section does not provide maximum information. Remember that 300 words are allowed for with this section, so please add as much information as possible here. Please note that this section is most considered important to decide for switching to your full text.

Thank you for pointing out the word count limitation in the Abstract. We agree that the Abstract serves as a critical component in helping readers decide whether to engage with the full manuscript.

In response to your comment, we have revised and expanded the Abstract to include additional methodological details, a more nuanced summary of the results (including specific odds ratios and subgroup findings), and a more comprehensive conclusion. The revised Abstract now contains 284 words, offering a fuller overview of our study’s rationale, methods, key findings, and clinical implications while remaining within the 300-word limit.

We believe the updated version improves clarity and informational value, as you suggested. Thank you again for your constructive feedback.

- With your Conclusions, please stick exclusively to your revised aims. Do not simply repeat your results here. Do not speculate. Instead, provide a reasonable and generalizable extension of your outcome.

Thank you for your valuable suggestion regarding the Conclusion section. In accordance with your advice, we have revised the Conclusion to better align with the stated aims of the study. Rather than repeating specific results or engaging in speculative interpretation, we now focus on providing a generalizable and objective summary of the study’s contribution.

Intro

- Please revise for sound spacebar use, see "(...) provider-side(1-3) and patient67 side(4-6) (...).". Again, stick to uniform Journal style, and revise thoroughly.

Thank you for pointing this out. We carefully reviewed the entire manuscript to ensure that all in-text references, citation brackets, and adjacent spacing are consistent with the PLOS ONE style guidelines. Specifically, we corrected missing or inappropriate spaces around citation numbers and terms (e.g., "provider-side (1–3) and patient-side (4–6)") to maintain uniform formatting throughout the manuscript.

- What about a sound rationale? Please revise carefully.

We appreciate this important comment. We have revised the final part of the Introduction to more clearly state the rationale for our study. Specifically, we now highlight that examining the visual structure of intake forms addresses an overlooked aspect of patient behavior that may be visible even before clinical interaction begins. We also explain how this perspective not only aims to fill an existing gap in dental risk assessment but may also provide practical insights to improve the design of future intake forms. We believe this addition strengthens the justification for our research focus and clarifies its potential clinical relevance.

- "Specifically, we asked whether the appearance of a patient's intake form -quantified (...)." What about a sound and valid null hypothesis?

Thank you for highlighting this important point. We have revised the Introduction to explicitly state the null hypothesis: that there is no association between the visual characteristics of the intake form and dental treatment prognosis. We believe this clarifies the testable premise of our study.

Meths

- Details or references must be provided with your "applicable institutional guidelines" and "regulatory requirements".

Thank you for this helpful comment. We have revised the Materials and Methods to explicitly state that the study was conducted in accordance with the Declaration of Helsinki and approved by the Fukuoka Dental College Ethics Committee (approval no. 541). This clarifies the applicable institutional and international ethical guidelines.

- "Python (v3.9) and the OpenCV (v4.5.5)" - more details must be given here. With ALL materials and methodologies (including statistical software), please use general names with your text, followed by (brand name; manufacturer, city, ST[ate - abbreviated, if US], country) in parentheses. Stick to semicolon. Remember that reproducibility must be ensured.

Thank you very much for your valuable feedback regarding the description of software and libraries for reproducibility. We have revised the manuscript to include full details according to your recommendation. Specifically, we have updated the Image Processing and Quantification of Questionnaire Filling Area in the Methods section as follows:

“To quantify the amount of writing on the patient questionnaires, all image processing tasks were performed using Python (Python Software Foundation; Wilmington, DE, USA) version 3.9 and the OpenCV library (OpenCV.org; Palo Alto, CA, USA) version 4.5.5. Each scanned or photographed questionnaire image was first converted to grayscale, followed by binarization using Otsu’s thresholding method (cv2.threshold). The number of black pixels was then calculated using cv2.countNonZero and subtracted from the total pixel count to estimate the amount of written content, represented as a proportion of all pixels. All images were processed under identical conditions. A detailed algorithm is provided in Supplementary1.”

- Do not use legal terms with your full text. Delete "Inc.", "Corp.", and so on.

Thank you for your comment. We have revised the Materials and Methods section to remove the term “Corp.” and have updated the company name to “IBM” in accordance with the journal style guidelines.

Results

- Please add statistical tests with your p values.

Thank you for your valuable feedback. We have now added the specific statistical tests used for each p-value throughout the Results section.

- Provide representative "photographed questionnaire images".

Thank you for your valuable comment. As requested, we have added representative scanned images of intake forms to the Supporting Information section. These images illustrate examples from each writing ratio group (Low, Mid, High), with all personal identifiers fully redacted to ensure confidentiality.

Please refer to Supplementary 2 for these representative images. We have also cited this figure in the Results section for clarity.

- The readers surely would like know what you mean when referring to "specifically the writing ratio".

Thank you for pointing this out. We have revised the manuscript to clarify the definition and calculation of the “writing ratio,” including its rationale and technical details. The revised text now reads:

“The writing ratio was defined as the percentage of black pixels (representing intake form itself and handwriting) relative to the total pixel count of the form. This writing ratio serves as a quantitative proxy for the visual complexity of the intake form — capturing how much content, whether typed or handwritten, appears on the form regardless of its semantic content.”

Disc

- Stick to H0 when starting this section.

Thank you for your suggestion. In response, we have revised the beginning of the Discussion section to clearly state the null hypothesis tested in this study. The revised text now reads:

“This study was designed to test the null hypothesis that the visual complexity of dental intake forms—quantified by the writing ratio—has no association with treatment outcomes. We focused on whether this structural characteristic, independent of the content itself, could serve as an early behavioral signal related to patient prognosis.”

This change clarifies the study’s hypothesis and provides a stronger conceptual foundation for interpreting our results. We have also revised the following paragraphs to improve logical flow and avoid repetition.

- What about the strengths of your study?

Thank you for pointing this out. In response, we have added a paragraph in the Discussion section to highlight the strengths of our study. The revised section includes the following:

“This study has several strengths. First, it introduces a novel and objective metric—the writing ratio—to quantify the visual complexity of dental intake forms, offering a unique lens through which to understand early patient engagement. Second, the use of image-based analysis allowed for standardized and reproducible quantification across a large sample. Third, by applying multivariable regression and subgroup analyses, we were able to control for key confounding factors such as age, sex, and diagnosis, thereby strengthening the robustness of our findings.”

We believe that explicitly stating these methodological and conceptual strengths provides a clearer context for evaluating the contribution of our work.

- What about the limitations?

Thank you for your comment. We have now added a paragraph explicitly addressing the limitations of our study in the Discussion section, following the paragraph on study strengths and before the concluding remarks. This new section outlines several important limitations, including the retrospective design, the reliance on a single-center dataset, the lack of detailed sociodemographic data, and the limited generalizability due to potential institutional or cultural factors. We believe this addition provides a more balanced interpretation of our findings and clarifies the scope of applicability for our results.

- What about answers to your teaching objective?

Thank you for this thoughtful suggestion. We have added a new paragraph at the end of the Discussion section before the strength and the limitation of this study to explicitly address the potential educational implications of our findings. Specifically, we propose that recognizing visual cues in intake forms—such as sparse or disorganized writing—may enhance clinical education by encouraging students and clinicians to consider early behavioral signs of disengagement or psychological burden. This addition highlights how the study’s findings can inform teaching strategies in dental education and promote more patient-centered care from the initial encounter.

Concl

- Again, with your Conclusions, please stick exclusively to your revised aims. Do not simply repeat your results here. Do not speculate. Instead, provide a reasonable and generalizable extension of your outcome.

Thank you for your valuable feedback regarding the Conclusions section. In response, we have revised the conclusion to align more closely with the revised aims of the study. Specifically, we focused on whether the structural characteristic of intake forms—quantified by the writing ratio—is associated with dental treatment outcomes. The revised conclusion avoids speculative statements and does not simply repeat the results. Instead, it presents a generalizable interpretation of our findings, emphasizing their potential relevance for clinical risk identification and intake form design. We hope this addresses your concern.

Refs

- Please adapt thoroughly to uniform Journal style.

Thank you for your comment. We have carefully revised all references to ensure they fully comply with the PLOS ONE journal style, including formatting of author names, journal titles, volume and issue numbers, page ranges, and DOI information where applicable.

Reviewer #2: Overall Comments:

The authors have done a good job overall, but the manuscript needs major revisions to improve clarity and flow. Some parts are confusing or feel disconnected. It would help to rewrite the paper with a clearer structure and storyline. The analysis section especially needs more explanation — it should clearly state what was analyzed and why.

We sincerely appreciate the reviewer’s constructive summary and thoughtful suggestions regarding the clarity, structure, and flow of the manuscript. In response, we have thoroughly revised the manuscript to improve coherence and readability throughout. We have reorganized several paragraphs, clarified the rationale and hypotheses in the Introduction, and improved the explanation of our statistical analysis in the Methods and Results sections. In particular, we now more clearly specify the variables included in the logistic regression model, explain the rationale for our categorizations (e.g., age and writing ratio). We believe these changes have significantly improved the manuscript’s structure and overall clarity, and we thank the reviewer for prompting these important revisions.

Specific Comments:

1. Lines 71–72: The sentence is unclear and should be rewritten for clarity.

Thank you for pointing this out. We have revised the sentence to improve clarity and ensure accurate expression of the interaction between provider-side and patient-side factors. The revised sentence reads:

"The interaction between provider-side and patient-side factors—particularly in terms of communication, trust, and the quality of the patient-dentist relationship—is also crucial for successful treatment outcomes."

2. Lines 82–93: These statements need to be supported by appropriate references.

Thank you for your comment. The paragraph in question was intended to describe clinically observed patterns rather than to cite specific empirical studies. We have revised the wording to clarify that these statements reflect theoretical considerations and clinical impressions, not findings from prior literature. We have also softened the language and added a qualifying clause to make this distinction explicit in the manuscript.

3. Line 110: Please clarify this line.

Thank you for pointing this out. We have revised the sentence to clarify that the preliminary analysis relied on OCR technology to extract text from scanned intake forms and count the number of characters as a proxy for the amount of writing. This method was later replaced with an image-based approach due to limitations in OCR accuracy.

4. Why was data extracted in 2022 for the 2016 fiscal year? Please explain the rationale.

Thank you for your question. Data from the 2016 fiscal year were extracted in 2022 to ensure that each case had a minimum follow-up period of over five years. This time interval was necessary to allow for sufficient observation of treatment progress, completion, or dropout over the long term.

5. Line 186: The statement that there are 288 patients in each group is incorrect. Please revise and ensure accuracy.

Thank you for pointing this out. We have corrected the inaccurate statement regarding group sizes. The number of patients in each writing ratio group has been updated to reflect the actual tertile-based distribution: Low (n = 271), Mid (n = 270), and High (n = 272), as now stated in the manuscript.

6. The mean age of participants is nearly identical across groups. Was this by chance, or was it an intentional aspect of the study design? Please clarify.

Thank you for your comment. The similarity in mean age across the Low, Mid, and High writing ratio groups was not intentional, but rather the result of natural distribution during the tertile-based classification. We did not stratify or match participants by age when grouping by writing ratio.

7. Writing Ratio (8.76%), please confirm whether this value relates to the high or medium writing ratio category. As written, it belongs to both.

Thank you for pointing out the ambiguity regarding the writing ratio category. We have revised the manuscript to clarify the cutoff values for each tertile. The Mid group now clearly refers to patients with a writing ratio ≥7.8900% and <8.7626%, and we have

---

## [Decision Letter · Decision Letter 1]

19 Aug 2025

Visual complexity of dental intake forms and its association with dental treatment outcomes: a retrospective cohort study

PONE-D-25-26414R1

Dear Dr. Umezaki,

We’re pleased to inform you that your manuscript has been judged scientifically suitable for publication and will be formally accepted for publication once it meets all outstanding technical requirements.

Kind regards,

Andrej M Kielbassa

Academic Editor

PLOS ONE

Additional Editor Comments (optional):

Reviewers' comments:

Reviewer's Responses to Questions

**Comments to the Author**

Reviewer #1: All comments have been addressed

Reviewer #2: All comments have been addressed

2. Is the manuscript technically sound, and do the data support the conclusions?

Reviewer #1: Yes

Reviewer #2: Yes

3. Has the statistical analysis been performed appropriately and rigorously?

Reviewer #1: Yes

Reviewer #2: Yes

4. Have the authors made all data underlying the findings in their manuscript fully available?

Reviewer #1: Yes

Reviewer #2: (No Response)

5. Is the manuscript presented in an intelligible fashion and written in standard English?

Reviewer #1: Yes

Reviewer #2: Yes

Reviewer #1: The Co-Authors have satisfyingly addressed all previous comments. This revised and re-submitted draft is considered ready to proceed.

Reviewer #2: I would like to thank the authors for addressing all my comments thoroughly. I will mark this manuscript as accepted; however, I strongly recommend that the authors add the age categories to Table 1 and include a legend with Figure 2, as described in their response to my comment.

**Do you want your identity to be public for this peer review?** For information about this choice, including consent withdrawal, please see our Privacy Policy

Reviewer #1: No

Reviewer #2: **Yes: ** Mona Abdelrehim

---

## [Editor Report · Acceptance letter]

PONE-D-25-26414R1

PLOS ONE

Dear Dr. Umezaki,

I'm pleased to inform you that your manuscript has been deemed suitable for publication in PLOS ONE. Congratulations! Your manuscript is now being handed over to our production team.

Kind regards,

on behalf of

Prof. Dr. med. dent. Dr. h. c. Andrej M Kielbassa

Academic Editor

PLOS ONE